# Higher Sodium Intake Assessed by 24 Hour Urinary Sodium Excretion Is Associated with Non-Alcoholic Fatty Liver Disease: The PREVEND Cohort Study

**DOI:** 10.3390/jcm8122157

**Published:** 2019-12-06

**Authors:** Eline H. van den Berg, Eke G. Gruppen, Hans Blokzijl, Stephan J.L. Bakker, Robin P.F. Dullaart

**Affiliations:** 1Department of Endocrinology, University of Groningen, University Medical Center Groningen, 9700RB Groningen, The Netherlands; e.g.gruppen@umcg.nl (E.G.G.);; 2Department of Gastroenterology and Hepatology, University of Groningen, University Medical Center Groningen, 9700RB Groningen, The Netherlands; h.blokzijl@umcg.nl; 3Department of Nephrology, University of Groningen, University Medical Center Groningen, 9700RB Groningen, The Netherlands; s.j.l.bakker@umcg.nl

**Keywords:** non-alcoholic fatty liver, sodium intake, insulin resistance, fatty liver index, hepatic steatosis index, HOMA-IR

## Abstract

A higher sodium intake is conceivably associated with insulin resistant conditions like obesity, but associations of non-alcoholic fatty liver disease (NAFLD) with a higher sodium intake determined by 24 hours (24 h) urine collections are still unclear. Dietary sodium intake was measured by sodium excretion in two complete consecutive 24 h urine collections in 6132 participants of the Prevention of Renal and Vascular End-Stage Disease (PREVEND) cohort. Fatty Liver Index (FLI) ≥60 and Hepatic Steatosis Index (HSI) >36 were used as proxies of suspected NAFLD. 1936 (31.6%) participants had an FLI ≥60, coinciding with the increased prevalence of type 2 diabetes (T2D), metabolic syndrome, hypertension and history of cardiovascular disease. Sodium intake was higher in participants with an FLI ≥60 (163.63 ± 61.81 mmol/24 h vs. 136.76 ± 50.90 mmol/24 h, *p* < 0.001), with increasing incidence in ascending quartile categories of sodium intake (*p* < 0.001). Multivariably, an FLI ≥60 was positively associated with a higher sodium intake when taking account for T2D, a positive cardiovascular history, hypertension, alcohol intake, smoking and medication use (odds ratio (OR) 1.54, 95% confidence interval (CI) 1.44–1.64, *p* < 0.001). Additional adjustment for the Homeostasis Model Assessment of Insulin Resistance (HOMA-IR) diminished this association (OR 1.30, 95% CI 1.21–1.41, *p* < 0.001). HSI >36 showed similar results. Associations remained essentially unaltered after adjustment for body surface area or waist/hip ratio. In conclusion, suspected NAFLD is a feature of higher sodium intake. Insulin resistance-related processes may contribute to the association of NAFLD with sodium intake.

## 1. Introduction

Non-alcoholic fatty liver disease (NAFLD) is characterized by hepatic steatosis in the absence of excessive alcohol use, and is emerging as the most common cause of chronic liver disease [1,2]. The spectrum of NAFLD ranges from simple steatosis to non-alcoholic steatohepatitis (NASH), fibrosis and eventually cirrhosis [1,3]. NAFLD coincides with obesity and insulin resistance, and is seen as the liver manifestation of the metabolic syndrome (MetS) [4]. NAFLD may in itself also increase the risk for the development of MetS and type 2 diabetes mellitus (T2D) [5,6]. NAFLD predisposes to plasma lipoprotein abnormalities, including elevations in apolipoprotein B-containing lipoproteins, as evidenced by elevations in very low-density lipoproteins (VLDL) and consequently in higher triglycerides, increased levels of low density lipoprotein (LDL) cholesterol, as well as in decreased levels of high density lipoprotein (HDL) cholesterol [2,7,8], which predisposes to atherosclerotic cardiovascular disease (CVD) [9]. The pathophysiological mechanisms underlying the development of NAFLD are not fully clarified, but multifactorial contributors including environmental factors (diet), central obesity, insulin resistance, alterations in gut microbiota and genetic factors are likely to play an important role [10].

In recent years, individual sodium intake is presumably increased, with the majority of extra sodium coming from diets filled-up with highly processed foods, particularly convenience foods [11]. High sodium intake is related to various metabolic disorders, such as obesity, insulin resistance, T2D, MetS, hypertension and CVD [11,12,13,14]. In turn, high sodium intake may also deleteriously influence metabolic diseases and is associated with expanding body fat and a diminished fat free mass [13,15]. Given increasingly documented associations of sodium intake with obesity [13,16,17], it is plausible to postulate that high sodium intake coincides with NAFLD and could even play a role in its pathogenesis. So far such a relationship has only been explored in two studies [14,18]. In these Asian studies, sodium intake was either estimated by dietary recall questionnaires [18] or by estimation from spot urine specimens [14]. Both studies found a positive relationship of NAFLD with estimated sodium intake [14,18]. However, the preferred method for assessing dietary salt intake is by measurement of sodium excretion in multiple 24 hour (24 h) urine collections, with other estimates being considered to be less accurate [19]. Thus, large-scale studies on the association of NAFLD with sodium intake measured by using multiple 24 h urine collections are still lacking.

Therefore, we initiated the present study to interrogate the impact of high sodium intake, assessed by using multiple 24 h urine collections, on NAFLD. We carried out a cross-sectional analysis among 6132 participants of in the Prevention of REnal and Vascular ENd-stage Disease (PREVEND) cohort study, comprising a large and well-characterized population from the North of The Netherlands.

## 2. Materials and Methods

### 2.1. Study Design and Population

This study was performed among participants of the Prevention of REnal and Vascular ENd-stage Disease (PREVEND) cohort study, a large prospective general population-based study that started in 1997 [20,21]. The PREVEND study was approved by the Medical Ethics Committee of the University Medical Center Groningen and is performed in accordance with Declaration of Helsinki guidelines [20,21]. Written informed consent was obtained from all participants. PREVEND was initiated to investigate cardiovascular and renal disease with a focus on albuminuria. All inhabitants (28 to 75 years old) of Groningen, the Netherlands were sent a questionnaire on demographics and cardiovascular morbidity, and were asked to supply urine specimens. Pregnant women, type 1 diabetic subjects and T2D subjects using insulin were excluded from participation. Participants with an urinary albumin concentration ≥10 mg/L were invited to our clinic together with randomly selected subjects with an urinary albumin concentration <10 mg/L. The initial study population of the PREVEND study comprised 8592 subjects who completed the total study screening program.

For the present study, we used data of participants who completed the second screening round in the PREVEND study (2001–2003; *n* = 6893), in which 24 h urine collections are available [22,23]. We excluded all subjects with missing values on urinary sodium excretion and subjects in which clinical and biochemical variables required to calculate the Fatty Liver Index (FLI), a proxy of NAFLD, were not available, leaving 6132 participants (Figure 1).

### 2.2. Measurements and Definitions

Procedures at examination, measurements and definitions in the PREVEND study are reported in detail previously [20,21,22,23,24]. Body mass index (BMI) was calculated as weight (kg) divided by height squared (meter). Body surface area (BSA) was calculated as the square root of height (cm) multiplied by weight (kg) divided by 3600. Waist circumference was measured as the smallest girth between rib cage and iliac crest [20]. The waist/hip ratio was determined as the waist circumference divided by the largest girth between waist and thigh. Blood pressure was assessed in a standardized fashion on the right arm in supine position, every minute for 10 min, with an automatic Dinamap XL Model 9300 series device. The mean of the last 2 recordings was used [23]. Lipid lowering drugs and other medications were not stopped prior to the clinical evaluation and blood sample collection. T2D was defined as a fasting glucose ≥7.0 mmol/L, a random glucose ≥11.1 mmol/L, self-report of a physician diagnosis or the use of glucose lowering drugs. Daily alcohol consumption was recorded with one alcoholic drink being assumed to contain 10 grams of alcohol. Smoking was categorized into current and never/former smokers. A positive cardiovascular history included: hospitalization for myocardial ischemia, obstructive coronary artery disease or revascularization procedures. Information on medication use was combined with information from a pharmacy-dispensing registry, which has complete information on drug usage of >95% of subjects in the PREVEND study [20,21].

Participants collected two consecutive 24 h urine specimens after thorough oral and written instruction. They were asked to avoid heavy exercise during the urine collection and instructed to postpone the urine collection in case of a urinary tract infection, menstruation or fever [22,23]. Urinary sodium concentration (in mmol/L) was multiplied by the urine volume in liters per 24 h to obtain a value in mmol/24 h. Results of two consecutive 24 h urine collections were averaged for analyses. Estimated glomerular filtration rate (eGFR) was calculated applying the combined creatinine cystatin C-based Chronic Kidney Disease Epidemiology Collaboration equation [25].

For the diagnosis of suspected NAFLD, we used the algorithm of the Fatty Liver Index (FLI) [26]. The FLI was calculated according to the following formula [26]:[e (0.953 × loge (triglycerides + 0.139 × BMI + 0.718 × loge (GGT) + 0.053 × waist circumference − 15.745)/[1 + e (0.953 × loge (triglycerides) + 0.139 × BMI + 0.718 × loge (GGT) + 0.053 × waist circumference − 15.745)] × 100(1)where GGT is gamma-glutamyltransferase.

The optimal cut-off value for the FLI is documented to be 60 with an accuracy of 84%, a sensitivity of 61% and a specificity of 86% for detecting suspected NAFLD as determined by ultrasonography [26]. Therefore, FLI ≥60 was used as proxy of NAFLD. The FLI is currently considered as one of the best-validated steatosis scores for larger scale screening studies [27]. Alternatively, we used the Hepatic Steatosis Index (HSI) [28]. The HSI is defined as follows:HSI = 8 × ALT/AST ratio + BMI (+2, if diabetes; +2, if female)(2)where ALT is alanine aminotransferase and AST is aspartate aminotransferase.

The cut-off value of the HSI for detecting suspected NAFLD is 36 [28]. In these equations, BMI is expressed in kg/m^2^, triglycerides are expressed in mmol/L, and GGT, ALT and AST are expressed in U/L.

The MetS was defined according to the revised National Cholesterol Education Program Adult Treatment Panel (NCEP-ATP) III criteria [29]. Participants were categorized with MetS when at least three out of five of the following criteria were present: waist circumference >102 cm for men and >88 cm for women; plasma triglycerides ≥1.7 mmol/L; HDL cholesterol <1.0 mmol/L for men and <1.3 mmol/L for women; hypertension (blood pressure ≥130/85 mm Hg or the use of antihypertensive medication); hyperglycemia (fasting glucose ≥5.6 mmol/L or the use of glucose lowering drugs). Homeostasis Model Assessment of Insulin Resistance (HOMA-IR) was calculated by:Fasting plasma insulin (mU/L) × fasting plasma glucose (mmol/L)/22.5(3)

### 2.3. Laboratory Methods

Laboratory methods are reported as described in detail previously [20,21,22,23,24]. Venous blood samples were drawn after an overnight fast while participants had rested for 15 min. Heparinized plasma and serum samples were obtained by centrifugation at 1400 × *g* for 15 min at 4 °C. Plasma and serum samples were stored at −80 °C until analysis. Glucose was measured directly after blood collection. Plasma total cholesterol, triglycerides and HDL cholesterol were measured as previously described [20,21,24]. Non-HDL cholesterol was calculated as the difference between total cholesterol and HDL cholesterol. LDL cholesterol was calculated by the Friedewald formula if triglycerides were <4.5 mmol/L [30]. Serum ALT and AST were measured using the standardized kinetic method with pyridoxal phosphate activation (Roche Modular P; Roche Diagnostics, Mannheim, Germany). Serum GGT was assayed by an enzymatic colorimetric method (Roche Modular P, Roche Diagnostics, Mannheim, Germany). Standardization of ALT, AST and GGT was performed according to International Federation of Clinical Chemistry guidelines [31,32,33]. High sensitivity C-reactive protein (hsCRP) was assayed by nephelometry. Serum creatinine was measured by an enzymatic method on a Roche Modular P analyzer (Roche Diagnostics, Mannheim, Germany). Serum cystatin C was measured by Gentian Cystatin C Immunoassay (Gentian AS, Moss, Norway) on a Modular analyzer (Roche Diagnostics, Indianapolis, IN, USA).

Urine samples collected at home were stored cold (4 °C) for a maximum of 4 days. After handing in the urine collections, specimens were stored at −20 °C. Urinary sodium was measured by indirect potentiometry with a MEGA clinical chemistry analyzer (Merck & Co., Inc., Kenilworth, NJ, USA). Urinary albumin excretion (UAE) was measured by nephelometry (Dade Behring Diagnostic, Marburg, Germany).

### 2.4. Statistical Analysis

IBM SPSS software (IBM Corp, version 23.0, Armonk, NY, USA) was used for data analysis. Results are expressed as mean ± standard deviation (SD), median with interquartile range (IQR) or as numbers (percentages). Normality of distribution was assessed and checked for skewness. HOMA-IR was log_e_-transformed for analysis to achieve an approximately normal distribution. Between group differences in variables were determined by unpaired *T*-tests or ANOVA (Analysis of Variance) test for normally distributed variables, Mann–Whitney U or Kruskal–Wallis test for non-normally distributed variables or by Chi-square tests for categorical variables where appropriate. Multivariable binary regression analyses were carried out to disclose the independent associations of urinary sodium excretion with an elevated FLI and HSI when taking account of clinical covariates and laboratory parameters. In multivariable analyses, urinary sodium excretion was expressed per 1 SD increment. Interactions were tested between 24 h sodium excretion and sex. Two-sided *p*-values < 0.05 were considered significant.

## 3. Results

### 3.1. Patient Characteristics

The study population consisted of 6132 subjects of whom 1936 (31.6%) were categorized with a FLI ≥60, as proxy of NAFLD. Table 1 shows the clinical characteristics and laboratory data of the participants according to FLI categorization. Subjects with a FLI ≥60 were older and more likely to be men (men 66.2% vs. women 33.8%), to be classified with MetS, T2D as well as hypertension and had a positive cardiovascular history more frequently. Consequently, subjects with FLI ≥60 used antihypertensive medication and glucose and lipid lowering drugs more frequently. BMI, BSA, waist circumference, waist/hip ratio, glucose, insulin, HOMA-IR, liver function tests, total cholesterol, non-HDL cholesterol, LDL cholesterol, triglycerides and creatinine were significantly higher in subjects with FLI ≥60, but eGFR and HDL cholesterol were lower. Sodium intake, determined by averaged 24 h urine sodium excretion was higher in participants with an FLI ≥60 (Table 1, 163.63 ± 61.81 mmol/24 h vs. 136.76 ± 50.90 mmol/24 h, *p* < 0.001).

Baseline characteristics categorized according to quartile categories of sodium intake are shown in Table 2. Suspected NAFLD, with either an FLI ≥60 or HSI >36 were both significantly higher in each ascending quartile category of urinary sodium excretion (*p* < 0.001). As the urinary sodium excretion increased, participants were more likely to be male, to be classified with MetS, T2D, had a higher BMI, BSA, waist circumference, waist/hip ratio and higher levels of glucose, insulin and HOMA-IR (all *p* < 0.001). Of note, hypertension and history of CVD were not significantly different between urinary sodium excretion quartiles.

### 3.2. Independent Associations of Suspected NAFLD with Sodium Intake

Multivariable binary regression analyses were subsequently performed in order to establish the independent associations of suspected NAFLD with urinary sodium excretion (Table 3 and Table 4). In age- and sex-adjusted analysis, FLI ≥60 was positively associated with an increased sodium excretion (expressed per 1 SD increment) (Table 3, Model 1, odds ratio (OR) 1.54, 95% confidence interval (CI) 1.45–1.64, *p* < 0.001). This positive association was also demonstrated after additional adjustment for presence of T2D, a positive history of CVD, hypertension, alcohol intake (≥10 and <10 g/day) and current smoking (Table 3, Model 2, OR 1.51, 95% CI 1.42–1.61, *p* < 0.001), and for eGFR, UAE, use of antihypertensive medication and glucose and lipid lowering drugs (Table 3, Model 3, OR 1.54, 95% CI 1.44–1.64, *p* < 0.001). Finally, after additional adjustment for HOMA-IR the association of FLI ≥ 60 with urinary sodium excretion was attenuated but remained present (Table 3, Model 4, OR 1.30, 95% CI 1.21–1.41, *p* < 0.001). In alternative analyses with an HSI >36 instead of FLI ≥60 (Table 4), essentially similar positive associations of HSI >36 with urinary sodium excretion were found (Table 4, Model 1–4). When the multivariably adjusted models (Table 3 and Table 4) were further adjusted for BSA, as a measure of body size, the positive association of increased sodium excretion remained present both with respect to an FLI ≥60 (OR 1.13, 95% CI 1.05–1.21, *p* < 0.001) and to an HSI >36 (OR 1.21, 95% CI 1.13–1.31, *p* < 0.001). Furthermore, when these models were further adjusted for waist/hip ratio as a measure of body fat distribution, the positive associations of increased sodium excretion remained present both with respect to a FLI ≥60 (OR 1.25, 95% CI 1.15–1.36, *p* < 0.001) and to an HSI > 36 (OR 1.37, 95% CI 1.27–1.48, *p* < 0.001).

The positive association of FLI ≥60 with urinary sodium excretion was present in women and men separately (Appendix A). Likewise, the association of HSI >36 with urinary sodium excretion was present in both women and men (Appendix A). However, neither the association of FLI ≥60 nor of HSI >36 was different between sexes (fully adjusted model including age; *p* interaction = 0.396 for FLI ≥ 60 and *p* interaction = 0.298 for HSI >36, respectively).

Sensitivity analyses in 3221 participants (Appendix A), after exclusion of subjects with alcohol intake ≥10 g/day, a positive history of CVD, presence of hypertension, impaired eGFR (<60 mL/min/1.73 m^2^), elevated UAE (>30 mg/24 h), use of antihypertensive medication, glucose lowering drugs and lipid lowering drugs, also showed positive associations of FLI ≥ 60 with urinary sodium excretion after adjustment for age, sex, T2D and current smoking (Appendix A, Models 1 and 2). After additional adjustment for HOMA-IR, the association of FLI ≥ 60 with urinary sodium excretion was again attenuated but remained significant (Appendix A, Model 3). Similar sensitivity analyses with HSI > 36 iterated these findings (Appendix A).

## 4. Discussion

In this large-scale, cross-sectional study in a predominantly Caucasian population we have demonstrated, to the best of our knowledge, for the first time a positive association of higher sodium intake, determined by two consecutive 24 h urinary sodium excretions, with suspected NAFLD. In our study we used an elevated FLI [26], and in alternative analyses an elevated HSI [28] as proxies of NAFLD, in keeping with international recommendations to use biomarkers to categorize subjects with suspected NAFLD in large-scale studies [27]. In multivariable regression analyses, accounting for various clinical variables, including BSA, as a measure of body size, eGFR and UAE, suspected NAFLD remained independently associated with a higher sodium intake. Interestingly, both for FLI and HSI, this association was attenuated after further adjustment for HOMA-IR. Taken together, the present study demonstrates that suspected NAFLD is positively associated with sodium intake. Furthermore, our current findings conceivably suggest that insulin resistance may represent a metabolic intermediate in explaining the relationship of higher sodium intake with NAFLD.

Two earlier cross-sectional studies have investigated the association of sodium intake with NAFLD [14,18]. Huh et al. described that Korean subjects recruited from the general population (Korea National Health and Nutrition Examination Surveys study) with suspected NAFLD had higher urinary sodium spot concentration values [14]. Choi et al. observed that higher sodium intake was associated with a greater prevalence of NAFLD among young and middle-aged asymptomatic Korean adults participating in a hospital-based cohort study. In this report, sodium intake was estimated by dietary recall questionnaires [18]. Very recently, perceived sodium intake determined by questionnaire was found to predict NAFLD development among Chinese people [34]. Our present results corroborate these previous studies. However, these surveys were performed in Asian populations with different dietary habits and sodium intake compared to Europeans [14,18], and used less reliable methods to estimate sodium intake compared to multiple 24 h urinary sodium excretion measurements [19]. The International Consortium for Quality Research on Dietary Sodium/Salt (TRUE) (with representative experts in hypertension, nutrition, statistics and dietary sodium) recommends that for a correct estimation of 24 h dietary sodium consumption, multiple complete 24 h urine samples are preferred to take account of short-term variations in dietary intake [19]. The hospital-based cohort could introduce another potential draw-back of the findings by Choi et al., which may limit extrapolation of their findings to the general population [18]. Hence, to date our study is the first study assessing the association of suspected NAFLD with high sodium intake, as determined by multiple 24 h urinary sodium excretion levels. Second, the current repost is the first to describe the positive association of high sodium intake in NAFLD in a predominantly Caucasian population.

The presently documented sodium intake amounted to 164 mmol/24 h and 137 mmol/24 h in subjects with and without an elevated FLI, respectively. In comparison, the World Health Organization currently recommends a salt intake of 5 grams per day, corresponding to 86 mmol of sodium per day [35], which is evidently lower than that documented in the present report. However, this recommendation was published later than the documentation of urinary sodium excretion used for the present report. In view of higher sodium intake in obese individuals [13,16,17], and the impact of obesity on NAFLD development [1], the here reported association of NAFLD with urinary sodium excretion is not unexpected. Furthermore, it has been suggested that the relationship of sodium intake with obesity is more pronounced in women [36], as presently documented for suspected NAFLD. While excessive sodium intake and its relation with obesity is to an important extent due to consumption of highly processed foods and increased total calorie intake [11,13,15], it should be noted that high sodium intake could directly impact on obesity development even independent from energy intake [37,38]. Chronic salt overload promotes adipocyte hypertrophy in rats [39,40]. Furthermore, increased leptin concentrations have been described in animal studies in response to high sodium intake [39]. Indeed, high sodium induces lipogenesis and inflammatory adipocytokine secretion in adipocytes, which may provide a possible mechanism for inflammatory adipogenesis in sodium-linked obesity [41], and in turn may also play a role in accelerated NAFLD development [42]. On the other hand, in experimental animal models, the inhibition of the renin-angiotensin-aldosterone system reduces the activation of stellate and Kupffer cells and reduces oxidative stress possibly leading to the improvement of NAFLD [43]. Furthermore, higher sodium intake is likely to downregulate the renin-angiotensin-aldosterone system [44], whereas low dietary sodium as well as exogenous angiotensin II suppress plasma adiponectin [45], an adipokine which has been proposed to protect against NAFLD development [46]. Thus, it seems unlikely that renin-angiotensin-aldosterone system-mediated effects are directly involved in the association of NAFLD with higher sodium intake.

Remarkably in our study, the association of a higher sodium intake and NAFLD was attenuated after adjusting for HOMA-IR, raising the possibility that insulin resistance may play a role in explaining the association of NAFLD with higher sodium intake. In rats, a high-salt diet may promote insulin resistance [47]. In humans, insulin resistance was independently and positively associated with a higher sodium intake [48,49]. The results of our cross-sectional study are in agreement with the hypothesis that a higher sodium intake, possibly via an effect on insulin resistance, promotes the development of NAFLD. Of further note, the association of an elevated FLI and HSI with a higher sodium intake remained present after adjustment for the BSA, and alternatively the waist/hip ratio. This would suggest that this association is at least in part independent of body size and body fat distribution, a potentially relevant finding in view of the reported relation of sodium consumption with adiposity [13,16,17]. Obviously, further research is needed to delineate the responsible pathogenic mechanisms more precisely.

The relationship of high sodium intake with cardiometabolic diseases is well established [11,36,50]. It is also evident that a higher sodium intake increases systemic blood pressure [44]. Interestingly, high blood pressure may also independently contribute to the development of NAFLD, even in the absence of obesity and MetS [51]. In turn, a bi-directional effect has been proposed where NAFLD represents an important risk factor for the development of hypertension, possibly involving insulin resistance [51]. From a clinical perspective our study supports the contention that it is important to take sodium intake into account when evaluating the adverse impact of NAFLD on the incidence of cardiovascular disease, hypertension and diabetes [6,9,52].

Our study has several strengths. Our sample size of over 6000 individuals enabled robust calculations on effect sizes and sufficiently powered subgroup and multivariable adjusted analyses. Additionally, the PREVEND population is well characterized, with extensive and standardized measurements [20,21]. On the other hand, a number of limitations and methodological aspects also need to be discerned. First, its cross-sectional design does not allow cause-effect relationships to be established with certainty, nor can we exclude the possibility of reversed causation. Second, an elevated FLI was chosen as a proxy of suspected NAFLD. Notably, the FLI and HSI do not translate in an absolute measure of hepatic fat accumulation. Thus over- and underestimation of suspected NAFLD could have occurred. Nonetheless, the FLI is considered to have sufficient accuracy for NAFLD assessment, and its current use is in line with international guidelines to apply biomarker scores in order to characterize NAFLD in larger-sized cohorts [26,27]. Moreover, the positive association of suspected NAFLD with sodium intake was confirmed using the HSI as an alternative algorithm for NAFLD categorization [28]. Performing liver ultrasound or liver biopsy for the diagnosis of NAFLD, was not feasible in the PREVEND cohort study. Third, we could not differentiate between simple hepatic steatosis and hepatic fibrosis; therefore, no relationship of hepatic fibrosis with sodium intake could be established. Fourth, to preclude collinearity with the FLI and/or HSI in the statistical analyses, variables making part of the equations (i.e., BMI and waist circumference) were not included in multivariable analyses. Instead we used BSA and waist/hip ratio in subsidiary analyses. Fifth, since alcohol intake and medical history were based on self-administered questionnaires, some misreporting by individuals cannot be excluded. However, considering the large number of subjects, this limitation is unlikely to have major effects on the interpretation of our results. Furthermore, the proportion of subjects using alcohol in excess of 30 gram per day in the PREVEND cohort is rather low, i.e., about 5.2% [53]. We adjusted for alcohol consumption and the association of an elevated FLI with sodium intake remained present in sensitivity analyses in which we excluded subjects with alcoholic intake ≥10 g/day. Furthermore, the PREVEND cohort is possibly enriched with people with micro-albuminuria. For this reason, we adjusted for eGFR and UAE in multivariable regression analysis and carried out a sensitivity analysis excluding subjects with impaired eGFR and elevated UAE. Reassuringly these analyses yielded similar positive and independent associations of suspected NAFLD with high sodium intake. Finally, no detailed information of diet composition is available in the PREVEND study. For this reason, we cannot exclude a contribution of unmeasured changes in diet components that are associated with a higher sodium intake in prevalent NAFLD, and the relationship of higher urinary sodium excretion and suspected NAFLD.

In conclusion, this study shows that suspected NAFLD is featured by a higher sodium intake. It seems conceivably that insulin resistance-related processes may explain in part the association of NAFLD with sodium intake.

## Figures and Tables

**Figure 1 jcm-08-02157-f001:**
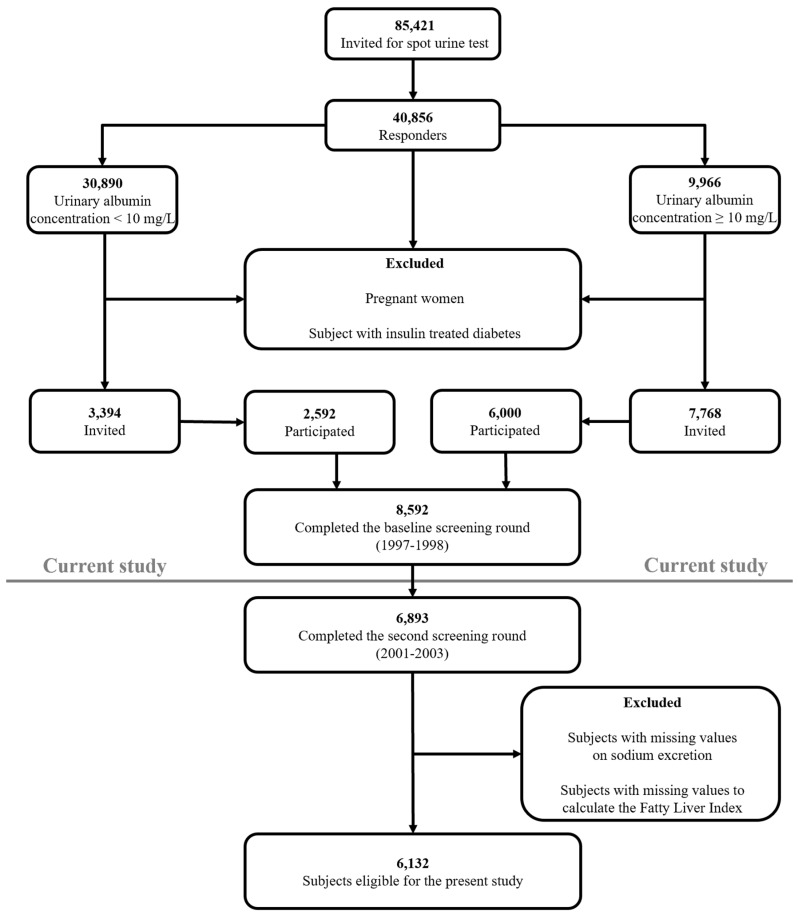
Flowchart of the study population.

**Table 1 jcm-08-02157-t001:** Baseline characteristics including averaged 24 h urinary sodium excretion (two collections) in 4196 subjects with a Fatty Liver Index (FLI) < 60 and 1936 subjects with an FLI ≥ 60.

	FLI < 60, *n* = 4196 (68.4%)	FLI ≥ 60, *n* = 1936 (31.6%)	*p* Value
Age (years), mean ± SD	52.2 ± 12.0	57.2 ± 11.4	<0.001
Sex			<0.001
Men, n (%)	1750 (41.7)	1282 (66.2)	
Women, n (%)	2446 (58.3)	654 (33.8)	
MetS, n (%)	372 (8.9)	1193 (61.7)	<0.001
Type 2 diabetes mellitus, n (%)	130 (3.1)	256 (13.2)	<0.001
History of cardiovascular disease, n (%)	192 (4.6)	194 (10.0)	<0.001
Hypertension, n (%)	1014 (24.2)	1050 (54.3)	<0.001
Current smokers, n (%)	1184 (28.2)	513 (26.5)	0.162
Alcohol ≥ 10 g/day, n (%)	139 (3.3)	117 (6.1)	<0.001
Antihypertensive medication, n (%)	613 (14.6)	703 (36.3)	<0.001
Glucose lowering drugs, n (%)	73 (1.7)	152 (7.9)	<0.001
Lipid lowering drugs, n (%)	273 (6.5)	307 (15.9)	<0.001
Systolic blood pressure (mm Hg), mean ± SD	121 ± 17	135 ± 18	<0.001
Diastolic blood pressure (mm Hg), mean ± SD	71 ± 9	77 ± 9	<0.001
BMI (kg/m^2^), mean ± SD	24.8 ± 2.9	30.9 ± 4.1	<0.001
BSA (m^2^), mean ± SD	1.87 ± 0.17	2.12 ± 0.18	<0.001
Waist circumference (cm), mean ± SD	86.0 ± 9.2	105.0 ± 9.2	<0.001
Waist/hip ratio, mean ± SD	0.87 ± 0.07	0.96 ± 0.07	<0.001
Glucose (mmol/L), mean ± SD	4.80 ± 0.86	5.48 ± 1.41	<0.001
Insulin (mU/L), median (IQR)	6.80 (5.1–9.3)	13.00 (9.5–19.0)	<0.001
HOMA-IR (mU mmol/L^2^/22.5), median (IQR)	0.36 (0.04–0.70)	1.10 (0.73–1.53)	<0.001
hsCRP (mg/L), median (IQR)	1.02 (0.49–2.32)	2.44 (1.23–4.47)	<0.001
ALT (U/L), median (IQR)	15 (12–20)	23 (17–32)	<0.001
AST (U/L), median (IQR)	22 (19–25)	25 (21–29)	<0.001
ALP (U/L), mean ± SD	65 ± 19	76 ± 23	<0.001
GGT (U/L), median (IQR)	19 (14–27)	40 (28–61)	<0.001
Total cholesterol (mmol/L), mean ± SD	5.32 ± 1.01	5.67 ± 1.06	<0.001
Non-HDL cholesterol (mmol/L), mean ± SD	3.98 ± 0.99	4.57 ± 1.02	<0.001
LDL cholesterol (mmol/L), mean ± SD	3.51 ± 0.92	3.74 ± 0.94	<0.001
HDL cholesterol (mmol/L), mean ± SD	1.34 ± 0.31	1.10 ± 0.24	<0.001
Triglycerides (mmol/L), median (IQR)	0.95 (0.72–1.26)	1.67 (1.28–2.20)	<0.001
Serum creatinine (umol/L), mean ± SD	82.92 ± 22.42	89.56 ± 19.23	<0.001
eGFR (mL/min/1.73 m^2^), mean ± SD	93.8 ± 16.4	86.5 ± 17.8	<0.001
Averaged 24 h urine excretion values			
Sodium excretion (mmol/24 h), mean ± SD	136.76 ± 50.90	163.63 ± 61.81	<0.001
UAE (mg/24 h), median (IQR)	7.44 (5.65–11.21)	10.04 (6.69–18.81)	<0.001

Data are given in number with percentages (%), mean ± standard deviation (SD) for normally distributed data or median with interquartile ranges (IQR) for non-normally distributed data. HOMA-IR was log_e_ transformed for analyses. LDL cholesterol was calculated by the Friedewald formula if triglycerides were <4.5 mmol/L in 6028 subjects. Abbreviations: 24 h, twenty-four hours; ALP, alkaline phosphatase; ALT, aminotransferase; AST, aspartate aminotransferase; BMI, body mass index; BSA, body surface area; FLI, Fatty Liver Index; eGFR, estimated glomerular filtration rate; GGT, gamma-glutamyltransferase; HOMA-IR, Homeostasis Model Assessment of Insulin Resistance; HDL, high density lipoproteins; hsCRP, high sensitivity C-reactive protein; LDL, low density lipoproteins; MetS, metabolic syndrome; T2D, type 2 diabetes mellitus; UAE, urinary albumin excretion.

**Table 2 jcm-08-02157-t002:** Baseline characteristics of the study population according to quartile categories of averaged 24 h urinary sodium excretion (two collections) in 6132 subjects.

	24 h Urinary Sodium Excretion				*p* Value
	Quartile 1	Quartile 2	Quartile 3	Quartile 4	
N (%)	1533 (25.0)	1532 (25.0)	1532 (25.0)	1533 (25.0)	
24 h Sodium excretion (mmol/day), mean ± SD	82.14 ± 18.02	122.92 ± 9.79	155.83 ± 10.08	220.06 ± 42.67	<0.001
Suspected NAFLD					
FLI ≥ 60, n (%)	329 (21.5)	386 (25.2)	505 (33.0)	715 (46.6)	<0.001
HSI > 36, n (%)	324 (21.1)	355 (23.2)	445 (29.0)	608 (39.7)	<0.001
Age (years), mean ± SD	54.9 ± 12.6	54.1 ± 12.1	53.0 ± 12.0	52.1 ± 11.3	<0.001
Sex					<0.001
Men, n (%)	466 (30.4)	632 (41.3)	822 (53.7)	1.112 (72.5)	
Women, n (%)	1067 (69.6)	900 (58.7)	710 (46.3)	421 (27.5)	
MetS, n (%)	333 (21.7)	344 (22.5)	375 (24.5)	512 (33.4)	<0.001
Type 2 diabetes mellitus, n (%)	82 (5.3)	89 (5.8)	98 (6.4)	117 (7.6)	0.007
History of cardiovascular disease, n (%)	96 (6.3)	90 (5.9)	100 (6.5)	100 (6.5)	0.605
Hypertension, n (%)	506 (33.0)	491 (32.1)	517 (33.7)	550 (35.9)	0.057
Current smokers, n (%)	493 (32.2)	427 (27.9)	385 (25.1)	391 (25.5)	<0.001
Alcohol ≥ 10 g/day, n (%)	66 (4.3)	65 (4.3)	52 (3.4)	73 (4.8)	0.810
Antihypertensive medication, n (%)	342 (22.3)	317 (20.7)	327 (21.3)	330 (21.5)	0.718
Glucose lowering drugs, n (%)	49 (3.2)	49 (3.2)	56 (3.7)	71 (4.6)	0.027
Lipid lowering drugs, n (%)	152 (9.9)	146 (9.5)	136 (8.9)	146 (9.5)	0.585
Systolic blood pressure (mm Hg), mean ± SD	124 ± 19	125 ± 20	125 ± 18	128 ± 17	<0.001
Diastolic blood pressure (mm Hg), mean ± SD	71 ± 9	73 ± 9	73 ± 9	74 ± 9	<0.001
BMI (kg/m^2^), mean ± SD	25.7 ± 4.1	26.1 ± 4.0	26.6 ± 4.3	28.1 ± 4.5	<0.001
BSA (m^2^), mean ± SD	1.86 ± 0.19	1.91 ± 0.19	1.97 ± 0.19	2.07 ± 0.20	<0.001
Waist circumference (cm), mean ± SD	87.9 ± 11.8	89.7 ± 11.9	91.8 ± 12.0	97.3 ± 12.9	<0.001
Waist/hip ratio, mean ± SD	0.87 ± 0.08	0.88 ± 0.08	0.90 ± 0.08	0.93 ± 0.08	<0.001
Glucose (mmol/L), mean ± SD	4.94 ± 1.07	4.96 ± 1.09	4.98 ± 1.11	5.12 ± 1.08	<0.001
Insulin (mU/L), median (IQR)	7.50 (5.3–10.9)	7.60 (5.5–11.0)	8.10 (5.8–11.8)	9.50 (6.6–14.3)	<0.001
HOMA-IR (mU mmol/L^2^/22.5), median (IQR)	0.49 (0.11–0.91)	0.46 (0.12–0.90)	0.53 (0.15–0.97)	0.72 (0.32–1.20)	<0.001
hsCRP (mg/L), median (IQR)	1.37 (0.65–3.07)	1.35 (0.58–3.09)	1.33 (0.61–2.87)	1.31 (0.63–3.08)	0.036
ALT (U/L), median (IQR)	15 (12–21)	16 (12–22)	17 (13–25)	20 (14–28)	<0.001
AST (U/L), median (IQR)	22 (19–26)	22 (19–26)	23 (20–26)	23 (20–27)	<0.001
ALP (U/L), mean ± SD	69 ± 21	69 ± 23	66 ± 18	69 ± 19	0.089
GGT (U/L), median (IQR)	21 (14–34)	21 (14–33)	23 (16–37)	27 (18–43)	<0.001
Total cholesterol (mmol/L), mean ± SD	5.45 ± 1.05	5.39 ± 1.03	5.37 ± 1.03	5.46 ± 1.04	0.372
Non-HDL cholesterol (mmol/L), mean ± SD	4.14 ± 1.04	4.11 ± 1.02	4.10 ± 1.04	4.26 ± 1.03	<0.001
LDL cholesterol (mmol/L), mean ± SD	3.59 ± 0.94	3.55 ± 0.92	3.54 ± 0.92	3.63 ± 0.94	0.079
HDL cholesterol (mmol/L), mean ± SD	1.31 ± 0.32	1.29 ± 0.32	1.27 ± 0.31	1.20 ± 0.29	<0.001
Triglycerides (mmol/L), median (IQR)	1.08 (0.79–1.45)	1.06 (0.75–1.55)	1.05 (0.80–1.54)	1.22 (0.86–1.75)	<0.001
Serum creatinine (umol/L), mean ± SD	82.77 ± 24.48	83.48 ± 17.41	85.78 ± 26.83	87.67 ± 15.93	<0.001
eGFR (ml/min/1.73 m^2^), mean ± SD	89.0 ± 17.1	91.0 ± 17.1	92.7 ± 17.3	94.2 ± 16.7	<0.001
UAE (mg/24 h), median (IQR)	6.90 (5.16–10.76)	7.77 (5.75–12.53)	8.21 (6.0–3.36)	9.24 (6.67–16.45)	<0.001

*p*-values represent *p* for trend. Data are given in number with percentages (%), mean ± standard deviation (SD) for normally distributed data or median with interquartile ranges (IQR) for non-normally distributed data. HOMA-IR was log_e_ transformed for analyses. LDL cholesterol was calculated by the Friedewald formula if triglycerides were <4.5 mmol/L (6028 subjects). Abbreviations: 24 h, twenty-four hours; ALP, alkaline phosphatase; ALT, aminotransferase; AST, aspartate aminotransferase; BMI, body mass index; BSA, body surface area; eGFR, estimated glomerular filtration rate; FLI, Fatty Liver Index; GGT, gamma-glutamyltransferase; HOMA-IR, Homeostasis Model Assessment of Insulin Resistance; HDL, high density lipoproteins; hsCRP, high sensitivity C-reactive protein; HSI, Hepatic Steatosis Index; LDL, low density lipoproteins; MetS, metabolic syndrome; T2D, type 2 diabetes mellitus; UAE, urinary albumin excretion.

**Table 3 jcm-08-02157-t003:** Multivariable regression analysis demonstrating the positive association of an elevated Fatty Liver Index (FLI ≥ 60) with averaged 24 h sodium excretion (two collections) after adjustment for clinical and laboratory covariates in 6132 subjects.

	Model 1		Model 2		Model 3		Model 4	
	OR (95% CI)	*p* Value	OR (95% CI)	*p* Value	OR (95% CI)	*p* Value	OR (95% CI)	*p* Value
Age (years)	1.04 (1.03–1.04)	<0.001	1.01 (1.01–1.02)	<0.001	1.00 (0.99–1.01)	0.665	0.99 (0.99–1.00)	0.131
Sex (men vs. women)	2.02 (1.79–2.28)	<0.001	2.02 (1.78–2.29)	<0.001	2.07 (1.82–2.37)	<0.001	2.48 (2.13–2.90)	<0.001
Sodium excretion per 24 h (1 SD increment)	1.54 (1.45–1.64)	<0.001	1.51 (1.42–1.61)	<0.001	1.54 (1.44–1.64)	<0.001	1.30 (1.21–1.41)	<0.001
Type 2 diabetes mellitus (yes/no)			3.13 (2.46–3.97)	<0.001	3.30 (2.26–4.82)	<0.001	0.48 (0.30–0.75)	0.001
History of cardiovascular disease (yes/no)			1.00 (0.79–1.27)	0972	0.77 (0.59–1.01)	0.057	0.71 (0.53–0.97)	0.031
Hypertension (yes/no)			2.94 (2.56–3.37)	<0.001	2.39 (1.99–2.88)	<0.001	2.04 (1.64–2.52)	<0.001
Alcohol intake (≥10 g/day vs. <10 g/day)			1.56 (1.18–2.06)	0.002	1.68 (1.26–2.24)	<0.001	2.23 (1.59–3.14)	<0.001
Current smoking (yes/no)			1.09 (0.95–1.25)	0.202	1.05 (0.91–1.21)	0.489	1.24 (1.05–1.46)	0.010
eGFR (mL/min/1.73 m^2^)					0.99 (0.98–0.99)	<0.001	0.99 (0.98–0.99)	<0.001
UAE (mg/24 h)					1.00 (1.00–1.00)	0.018	1.00 (1.00–1.00)	0.104
Use of antihypertensive medication (yes/no)					1.22 (0.99–1.50)	0.063	1.01 (0.80–1.29)	0.909
Use of glucose lowering drugs (yes/no)					0.89 (0.55–1.45)	0.647	1.83 (1.05–3.20)	0.033
Use of lipid lowering drugs (yes/no)					1.50 (1.20–1.87)	<0.001	1.25 (0.98–1.61)	0.077
HOMA-IR (mU mmol/L^2^/22.5)							9.45 (8.10–11.01)	<0.001

OR, odds ratio; 95% CI, 95% confidence intervals. OR is given per 1 SD increase for urinary sodium excretion. 1 SD change in urinary sodium excretion corresponds to 55.99 mmol sodium per day. HOMA-IR was log_e_ transformed for analyses. Abbreviations: 24 h, twenty-four hours; eGFR, estimated glomerular filtration rate; FLI, Fatty Liver Index; HOMA-IR, Homeostasis Model Assessment of Insulin Resistance; UAE; urinary albumin excretion. Model 1: adjusted for age and sex. Model 2: adjusted for age, sex, presence of type 2 diabetes, history of cardiovascular disease, presence of hypertension, alcohol intake and current smoking. Model 3: adjusted for age, sex, presence of type 2 diabetes, history of cardiovascular disease, presence of hypertension, alcohol intake, current smoking, estimated glomerular filtration rate, urinary albumin excretion, use of antihypertensive medication, glucose lowering drugs and lipid lowering drugs. Model 4: adjusted for age, sex, presence of type 2 diabetes, history of cardiovascular disease, presence of hypertension, alcohol intake, current smoking, estimated glomerular filtration rate, urinary albumin excretion, use of antihypertensive medication, glucose lowering drugs and lipid lowering drugs and HOMA-IR.

**Table 4 jcm-08-02157-t004:** Multivariable regression analysis demonstrating the positive association of an elevated Hepatic Steatosis Index (HSI > 36) with averaged 24 h sodium excretion (two collections) after adjustment for clinical and laboratory covariates in 6132 subjects.

	Model 1		Model 2		Model 3		Model 4	
	OR (95% CI)	*p* Value	OR (95% CI)	*p* Value	OR (95% CI)	*p* Value	OR (95% CI)	*p* Value
Age (years)	1.03 (1.02–1.03)	<0.001	1.00 (1.00–1.01)	0.906	0.99 (0.98–1.00)	0.021	0.99 (0.98–1.00)	0.003
Sex (men vs. women)	0.62 (0.55–0.70)	<0.001	0.59 (0.52–0.67)	<0.001	0.61 (0.53–0.70)	<0.001	0.57 (0.49–0.65)	<0.001
Sodium excretion per 24 h (1 SD increment)	1.63 (1.54–1.74)	<0.001	1.59 (1.49–1.70)	<0.001	1.59 (1.49–1.70)	<0.001	1.40 (1.31–1.51)	<0.001
Type 2 diabetes mellitus (yes/no)			5.01 (3.95–6.35)	<0.001	5.36 (3.69–7.79)	<0.001	1.69 (1.12–2.56)	0013
History of cardiovascular disease (yes/no)			0.82 (0.64–1.05)	0.109	0.66 (0.50–0.88)	0.004	0.62 (0.46–0.84)	0002
Hypertension (yes/no)			2.41 (2.09–2.77)	<0.001	2.07 (1.71–2.50)	<0.001	1.77 (1.44–2.16)	<0.001
Alcohol intake (≥10 g/day vs. <10 g/day)			1.28 (0.95–1.73)	0.106	1.34 (0.99–1.82)	0.060	1.44 (1.04–2.01)	0.029
Current smoking (yes/no)			0.73 (0.63–0.83)	<0.001	0.71 (0.61–0.82)	<0.001	0.74 (0.64–0.87)	<0.001
eGFR (mL/min/1.73 m^2^)					0.99 (0.99–1.00)	0.001	1.00 (0.99–1.00)	0.198
UAE (mg/24 h)					1.00 (1.00–1.00)	0.800	1.00 (1.00–1.00)	0.730
Use of antihypertensive medication (yes/no)					1.18 (0.96–1.45)	0.120	1.01 (0.81–1.26)	0.942
Use of glucose lowering drugs (yes/no)					0.83 (0.52–1.34)	0.450	1.30 (0.77–2.18)	0.325
Use of lipid lowering drugs (yes/no)					1.42 (1.13–1.77)	0.002	1.25 (0.98–1.58)	0.068
HOMA-IR (mU mmol/L^2^/22.5)							4.04 (3.56–4.57)	<0.001

OR, odds ratio; 95% CI, 95% confidence intervals. OR is given per 1 SD increase for urinary sodium excretion. 1 SD change in urinary sodium excretion corresponds to 55.99 mmol sodium per day. HOMA-IR was log_e_ transformed for analyses. Abbreviations: 24 h, twenty-four hours; eGFR, estimated glomerular filtration rate; HOMA-IR, Homeostasis Model Assessment of Insulin Resistance; HSI, Hepatic Steatosis Index; UAE; urinary albumin excretion. Model 1: adjusted for age and sex. Model 2: adjusted for age, sex, presence of type 2 diabetes, history of cardiovascular disease, presence of hypertension, alcohol intake and current smoking. Model 3: adjusted for age, sex, presence of type 2 diabetes, history of cardiovascular disease, presence of hypertension, alcohol intake, current smoking, estimated glomerular filtration rate, urinary albumin excretion, use of antihypertensive medication, glucose lowering drugs and lipid lowering drugs. Model 4: adjusted for age, sex, presence of type 2 diabetes, history of cardiovascular disease, presence of hypertension, alcohol intake, current smoking, estimated glomerular filtration rate, urinary albumin excretion, use of antihypertensive medication, glucose lowering drugs and lipid lowering drugs and HOMA-IR.

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
