# Peer review of "Higher Sodium Intake Assessed by 24 Hour Urinary Sodium Excretion Is Associated with Non-Alcoholic Fatty Liver Disease: The PREVEND Cohort Study"

_jcm, 2019, doi:10.3390/jcm8122157_

Round 1
Reviewer 1 Report
This study aimed to investigate the association between higher sodium intake and NAFLD, using a large cohort of patients and, therefore, giving significant strength to the study. Although the novelty is limited, the data are robust and well presented. I have no specific critics.
Author Response
This study aimed to investigate the association between higher sodium intake and NAFLD, using a large cohort of patients and, therefore, giving significant strength to the study. Although the novelty is limited, the data are robust and well presented. I have no specific critics.
Response: We thank the reviewer for his/her positive comments.
Reviewer 2 Report
In this manuscript titled, " Higher sodium intake assessed by 24-hour urinary sodium excretion is associated with non-alcoholic fatty liver disease: The PREVEND cohort study ", Eline H. van den Berg et al., authors investigated the impact of high sodium intake, assessed by using multiple 24-h urine collections, on Non-alcoholic fatty liver disease (NAFLD). The authors revealed that suspected NAFLD is featured by higher sodium intake. Insulin resistance-related processes may contribute to the association of NAFLD with sodium intake. Overall, the manuscript is written clearly. For the study, the presented data are quite sufficient. However, the manuscript appears preliminary.
Abbreviations must be defined at first mention in the abstract, and again at first mention in the main manuscript text, thereafter the abbreviation/acronym should be used. For figure 1 you should improve the quality of the panels of figure 1, you may also try to enlarge every panel of this figure for a good observation.
Author Response
In this manuscript titled, " Higher sodium intake assessed by 24-hour urinary sodium excretion is associated with non-alcoholic fatty liver disease: The PREVEND cohort study ", Eline H. van den Berg et al., authors investigated the impact of high sodium intake, assessed by using multiple 24-h urine collections, on Non-alcoholic fatty liver disease (NAFLD). The authors revealed that suspected NAFLD is featured by higher sodium intake. Insulin resistance-related processes may contribute to the association of NAFLD with sodium intake. Overall, the manuscript is written clearly. For the study, the presented data are quite sufficient. However, the manuscript appears preliminary.
Response: We appreciate the positive comments of the reviewer on our study. We agree that our present cross-sectional results are preliminary, in so far that the mechanisms responsible for the interplay between NAFLD, sodium consumption and insulin resistance, as well as the longitudinal association between sodium intake and future NAFLD development, need to be delineated more precisely in future studies.
Abbreviations must be defined at first mention in the abstract, and again at first mention in the main manuscript text, thereafter the abbreviation/acronym should be used. For figure 1 you should improve the quality of the panels of figure 1, you may also try to enlarge every panel of this figure for a good observation.
Response: Thank you for this comments. In the revised manuscript we have re-defined the abbreviations first mentioned in the abstract and the main text. Also, we have improved and enlarged Figure 1 in the revised manuscript.
Reviewer 3 Report
In this work, Van den Berg and colleagues demonstrated a positive association of higher sodium excretions with suspected NAFLD in a large cohort study. Several studies treated this topic before, but they were focused in the Asian population, which displayed different food habits. Overall, this paper is well written and the data showed are clear.
Author Response
Response: We thank the reviewer for his/her positive review of our manuscript.